# From Sport Psychology to Action Philosophy: Immanuel Kant and the Case of Video Assistant Referees

**DOI:** 10.3390/bs14040291

**Published:** 2024-04-01

**Authors:** Yair Galily

**Affiliations:** 1Sport, Media and Society (SMS) Research Lab, Sammy Ofer School of Communications, Reichman University (IDC) Herzliya, Herzliya 46150, Israel; yg2@runi.ac.il; 2Baruch Ivcher School of Psychology, Reichman University (IDC) Herzliya, Herzliya 46150, Israel

**Keywords:** soccer, football, decision making, ethics, emotions, VAR

## Abstract

The implementation of Video Assistant Referees (VARs) in 2018 has had a significant impact on the multi-billion-dollar soccer industry. As *the* most popular and watched sport globally, soccer’s financial stakes are high, with clubs, leagues, broadcasters, sponsors, and fans heavily invested in the game. The ongoing debate surrounding the VAR system brings to light the intricate balance between preserving the authenticity of football (soccer) and harnessing technology to improve accuracy. It is crucial to strike the right equilibrium in order to uphold football’s metaphorical power and sustain the timeless joy it has brought to fans throughout generations. In this context, Immanuel Kant’s philosophy can offer valuable insights into the utilization of VARs in soccer. According to Kantian ethics, using VARs can be justified if it serves to enhance fairness and accuracy, aligning with the moral duties of referees. Nevertheless, it is important to consider the potential dehumanizing effects and the necessity of preserving the value of human judgment in the game. Therefore, this paper aims to explore in-depth the intricate dynamics that arise when technology is integrated into traditional practices, emphasizing the significance of critical reflection on the implications of such advancements.

Sapere aude! Have the courage to use your own understanding (*Immanuel Kant*, 1784)

## 1. Kick-Off: The Essence of Football

Football, also known as soccer, is frequently considered a metaphor for life and holds a profound significance in the hearts of its passionate followers. In Spanish, a ball is referred to in the feminine gender, symbolizing its elusive yet highly coveted nature. South American culture, in particular, cherishes the ball, seeing it as a concept that evokes love, joy, and celebration—an emotional experience. The game itself serves as a parable for life’s triumphs and tribulations, reflecting our unrealized aspirations, moments of intense excitement, and the gratitude we feel when witnessing the seemingly impossible being achieved. When a goal is scored, it transcends time, suspending true joy and sadness until its confirmation and official recognition. Only then are we encouraged to express our elation or sorrow. However, with the introduction of VARs (Video Assistant Referees), this spontaneous movement of the mind is replaced by a mere sigh of relief. Consequently, the essence of the football experience we crave and seek has allegedly been lost [1]

Football undeniably mirrors life, as argued by George Orwell [2]. It allows for a refined conflict, reminiscent of war, within a protected framework and without bloodshed. If we aim to instill in football fans a model from which they can emerge with an optimistic outlook, it becomes crucial to provide them with the best means to test the quality of both teams and minimize cases where games are decided by incorrect referee decisions. Improving the quality and accuracy of refereeing in football could also have an impact on our expectations of referees presiding over real-life courts.

In terms of legal interpretation, every law can be approached from two angles: the intention of the legislature (referred to by proponents of the judicial coup as the “subjective purpose”) and the proper purpose of the law as perceived by the judge (oddly labeled as the “objective purpose”). Recently, Shoval [1], similar to other scholars, has aligned with the prevailing view in today’s courts, which asserts that the subjective purpose should take a backseat to the objective purpose, with the referee striving for “substantial football.” Conversely, VARs are based on the opposite perception.

The judicial revolution is rooted in a postmodern conception that rejects the existence of absolute truth and favors the idea of multiple narratives, each holding equal weight. The judge’s role, according to this view, is to select the “reasonable” or “enlightened” narrative, even if it contradicts the truth and the language of the law. This approach grants the judge omnipotence, subordinating reality to the judge’s worldview. In contrast, VARs promote the belief in absolute reality, emphasizing that the judge’s duty is to uncover reality as it is and make rulings solely in accordance with it. In this context, the judge is not omnipotent but rather an official limited to the laws established by the legislature.

Shoval [1] also expresses another postmodern perception, suggesting that excitement is the ultimate purpose of the game, paralleling it with the purpose of life itself. In his article, he writes, “In the great life parable of the game, the lack of clarity about the referee’s arbitrary decision gave us hope that maybe one day we too will be good.” This sentiment implies a conscious preference for injustice, solely to allow the disadvantaged side to hold onto the hope that the judge will make an error.

Some (see, for example [3,4]) would argue that the integration of VARs into football challenges common sense and alters the essence of the game. It introduces a new paradigm that clashes with traditional notions of football’s purpose, fairness, and the symbolic journey it represents.

Thus, the aim of the current paper is, with the help of Kant’s philosophy, to shed light on the ongoing debate surrounding VARs, which highlights the complex interplay between preserving the authenticity of the game and leveraging technology to enhance accuracy. Ultimately, striking the right balance is crucial to ensuring that football retains its metaphoric power and continues to evoke the raw and childlike happiness that has captivated fans for generations.

## 2. Video Assistant Referees (VARs) in Soccer: Enhancing Fairness and Accuracy?

The VAR is widely regarded as a major milestone in the field of football officiating, comparable to the introduction of yellow and red cards in the 1970s. Despite facing initial resistance from FIFA, the central governing body of football, the VAR was eventually embraced and officially adopted in 2018, following its earlier implementation by certain national leagues in the preceding year [4].

The Video Assistant Referee (VAR) is an innovative technology introduced in soccer to aid match officials in making more accurate and fair decisions during games. It involves a team of video assistant referees who review incidents and provide additional information to the on-field referee to help them make informed judgments. While VARs have generated both praise and criticism, their primary objective is to reduce errors and promote fairness within the game [5,6,7,8].

The original decision by the referee will *only* be changed if the video review clearly shows a mistake. Research (see, for example, [9]) indicates that around 26%(!) of referee (including linesmen) decisions per game are errors, highlighting the need for VARs to correct these mistakes and improve decision-making accuracy. Psychological factors like the Crowd Effect, Home Advantage Effect, Player Body Type Phenomenon, ‘Dirty Team’ Effect, and National Bias can influence a referee’s objectivity and decision-making process. The VAR aims to reduce biases by providing an objective measure for decision-making, helping referees make better judgments without the influence of subconscious bias. Indeed, decision-making (DM) is a critical aspect of soccer refereeing. Referees make numerous repetitive decisions, many of which are dependent on appropriate field location and effective interaction with their assistants. Considering the complexity of the sequential DM process, it is not surprising that referees exhibit high decision error rates. In the realm of officiating, various models underscore the significance of perception in shaping referee decision-making. One such model, as presented by MacMahon et al. [10], outlines perception as the initial phase, succeeded by categorization and information integration, culminating in the decision-making process. Similarly, the model developed by Samuel et al. [11], tailored specifically for football referees, underscores the importance of both the direction of movement and the focal points in decision execution. According to this model, perception involves multiple senses, such as crowd noise, which can impact a referee’s performance. Nevertheless, visual perception remains paramount for referees. Research indicates that elite performance in both referees and athletes is characterized by sport-specific visual behaviors. The proposition suggests that enhanced gaze behavior results from optimal information reduction, with elite performers demonstrating a superior ability to differentiate between relevant and irrelevant information sources and allocate their gaze accordingly toward the most pertinent sources of information. While there is no official body or organization that tracks and publishes comprehensive data on the number or percentage of decisions overturned by VAR across all competitions that use it. However, in the English Premier League, VARs have significantly impacted various clubs in different seasons. In the 2023-24 season, there have been a total of 67 overturns, 22 leading to goals, 30 leading to disallowed goals, and 20 penalties awarded [12]. 

Indeed, the implementation of VARs has had a significant impact on the multi-billion-dollar soccer industry. As the most popular and watched sport globally, soccer’s financial stakes are high, with clubs, leagues, broadcasters, sponsors, and fans heavily invested in the game.

In addition, VARs presumably enhance the credibility of the sport and contribute to a more engaging and satisfying experience for fans. They help maintain the trust and loyalty of supporters, who invest their time, money, and emotions into following their favorite teams and players.

Undeniably, soccer referees face the demanding task of making critical judgments in dynamic game situations, frequently characterized by rapid movement, multiple players, and a multitude of other factors. Despite the challenges posed by limited visibility [13], referees must adeptly navigate these complexities to officiate effectively.

The introduction of VAR in soccer has aimed to address certain limitations of the traditional refereeing system. In fast-paced and high-stakes matches, referees can sometimes struggle to observe and assess every critical moment accurately. VARs offer an additional layer of scrutiny, allowing officials to review contentious incidents, such as goals, penalties, red card offenses, and cases of mistaken identity. By utilizing video replays and multiple camera angles, the VAR aims to minimize human errors and ensure that crucial decisions are based on more comprehensive information.

The implementation of VARs follows a structured process. When an incident occurs, the on-field referee has the option to request a review from the video assistant referee team. The VAR team then analyzes the footage and provides their input to the referee through a communication system. The referee can choose to review the incident themselves on a pitchside monitor or rely on the guidance and advice of the VAR team. Ultimately, the final decision rests with the on-field referee, who maintains the authority to accept or reject the VAR’s recommendation [14,15].

The introduction of VARs has had several notable impacts on the game. Firstly, it has increased the overall accuracy of refereeing decisions. The use of technology has enabled officials to rectify clear and obvious errors that may have otherwise affected the outcome of a match. Additionally, VARs have contributed to a fairer playing field by reducing instances of diving, simulation, and other forms of unsporting behavior, as players are aware that their actions can be subject to review.

However, despite its intentions, VAR has not been without controversy. Critics argue that the technology disrupts the flow and spontaneity of the game, leading to extended interruptions and delays. The interpretation of VAR incidents can also be subjective, leaving room for differing opinions and potential controversies. Some believe that the reliance on technology undermines the authority and judgment of the on-field referee, shifting the decision-making power to the VAR team.

Efforts are being made to address these concerns and refine the implementation of VARs. Leagues and football governing bodies continue to refine the protocols, aiming to minimize disruptions and provide clearer guidelines for VAR decisions. Referees are receiving extensive training to ensure consistent and effective use of the technology. Additionally, transparency in communication and involving fans in the decision-making process through video screens and audio announcements at stadiums help enhance understanding and acceptance of VAR.

The VAR is an ongoing development in soccer, with its effectiveness and impact being closely monitored and evaluated. The ultimate goal remains to strike a balance between maintaining the flow and spirit of the game while ensuring fair and accurate outcomes. As technology continues to advance, VAR is likely to evolve, potentially integrating advancements such as automated offside technology and goal-line detection to further enhance the precision of decision-making.

In summary, the introduction of VARs in soccer represents a significant step toward improving the fairness and accuracy of refereeing decisions. While there are debates surrounding its implementation and impact on the game, the VAR’s objective is to minimize errors and ensure that pivotal moments are determined based on comprehensive information. As the technology continues to evolve and guidelines are refined, the aim is to strike a balance that upholds the integrity of the game while leveraging technology to enhance the overall officiating process.

## 3. Half-Time: Issues and Controversies

The employment of Video Assistant Referees (VAR) in soccer has not been without controversy. While the technology aims to enhance fairness and accuracy, its application has sparked debates and criticism within the football community. Here are some of the main points of contention associated with VAR:

One of the primary concerns raised by critics is the impact of VARs on the flow and tempo of the game, leading to delays and disruptions. VAR reviews can lead to lengthy interruptions, disrupting the natural rhythm and excitement of the match. Fans and players have expressed frustration with the extended stoppages, which can dampen the overall spectator experience [14].

In addition, VAR decisions can sometimes be subjective, leading to differing interpretations and debates. While the technology provides access to multiple camera angles and replays, there is still room for interpretation and judgment when reviewing incidents. This subjectivity has resulted in instances where controversial decisions have been made despite the availability of video evidence.

For example, offside decisions with VAR have generated significant disagreement. The technology allows for precise measurements and freeze-frame analysis, which has led to marginal offside calls. Critics argue that such decisions go against the spirit of the game, as players can be penalized for seemingly insignificant infractions that are difficult to perceive in real time. At the same time, VAR decisions have sometimes lacked consistency, with similar incidents resulting in different outcomes depending on the interpretation of the VAR officials. This inconsistency has led to confusion and a sense of unfairness, as identical situations can be judged differently from one match to another. It has also raised questions about the standardization and application of VAR across different leagues and competitions [16].

Subsequently, VAR decisions often lack transparency and clarity, leading to confusion among players, coaches, and fans. The communication between the on-field referee and the VAR team is typically not visible or audible to spectators in the stadium, which can leave them in the dark regarding the reasoning behind decisions. This lack of transparency has contributed to a sense of frustration and skepticism among stakeholders.

In addition, some critics argue that VARs have resulted in an overreliance on technology and diminished the authority of the on-field referee. The decision-making process has shifted to the VAR team, potentially undermining the referee’s judgment and experience. This shift in power dynamics has sparked debates about the role of technology versus the role of human decision-making in the game.

Ultimately, VARs have had an impact on the emotional aspect of the game. Moments of celebration or disappointment after a goal can be muted by the need to wait for a VAR review before confirming the validity of the goal. The emotional rollercoaster of experiencing a goal is altered, and some argue that it detracts from the spontaneity and raw emotion that makes football so captivating. In that respect, VARs have also brought attention to the issue of marginal decisions, where the technology’s precision can lead to decisions that seem overly technical or pedantic. This includes situations where a player’s toe or shoulder is ruled offside by a matter of centimeters. Critics argue that such decisions undermine the spirit of the game and create frustration among fans, players, and coaches.

The VAR has also altered the dynamic of goal celebrations (see, for example [17]). Fans, players, and coaches often delay their celebrations until the VAR review confirms the validity of a goal. This pause diminishes the immediate joy and spontaneity associated with scoring a goal, as there is always a lingering uncertainty until the final decision is made. This has had an impact on the atmosphere in stadiums, where celebrations have become more tentative and subdued.

While VAR was intended to minimize human errors, it has not completely eliminated subjectivity from refereeing decisions. The final judgment still lies with the on-field referee, who can choose to accept or reject the recommendations of the VAR team. This subjective element has led to disagreements and debates about the extent to which VAR has truly improved the fairness and accuracy of decision-making.

Critics also argue that VAR disrupts the natural pace and intensity of soccer matches. The frequent breaks for VAR reviews can break the momentum and rhythm of the game, reducing the overall excitement and flow. This is particularly evident in instances where goals are disallowed or penalties are awarded after a VAR intervention, as it can drastically change the course of a match.

In addition, VARs have affected the fan experience and engagement with the game. The uncertainty and delays associated with VAR reviews have led to frustration among fans in the stadium, who may not have access to the same replays and information as those watching on television. This has raised concerns about the potential alienation of match-going fans and the overall enjoyment of the live match experience.

Despite the implementation of VAR, accusations of referee bias or favoritism persist. Some fans and teams believe that VAR decisions are influenced by factors such as the reputation of players or teams involved, leading to claims of unfair treatment. These perceptions of bias can further contribute to the controversy and skepticism surrounding VAR.

The controversy surrounding VAR continues to be a topic of discussion and debate within the soccer community. While the technology has undoubtedly introduced new challenges and raised questions about the game’s traditions, ongoing efforts are being made to refine its implementation and address the concerns raised by its critics.

## 4. Leveraging Technology for Accuracy and Communication Challenges

Efforts are being made to address these controversies and improve the implementation of VAR. Football governing bodies and leagues are continually reviewing and adjusting the protocols to minimize disruptions, increase transparency, and enhance the understanding of VAR decisions among fans, players, and officials. The aim is to strike a balance between leveraging technology for accuracy while preserving the essence and excitement of the game.

Communication issues between VAR referees and on-field officials have been a topic of discussion in the implementation of VAR in soccer. Here are some common challenges and considerations related to communication between VAR referees and on-field officials:

Communication can also be hindered by technical difficulties or malfunctions of the communication system used by VAR referees and on-field officials. Issues with audio clarity, delays in transmission, or equipment malfunctions can disrupt effective communication and lead to confusion or delays in decision-making.

In international competitions where referees and VAR officials may come from different countries, language barriers can pose challenges to effective communication. Clear and efficient communication is vital for accurate decision-making, and language differences can potentially lead to misunderstandings or delays in conveying information.

The limited time available for communication between VAR referees and on-field officials can sometimes impede effective information exchange. VAR reviews require swift decision-making to minimize disruptions to the flow of the game. Therefore, it is essential for VAR referees to quickly and concisely convey relevant information to the on-field officials to facilitate prompt decision-making.

VAR referees often communicate with on-field officials through signals or gestures, such as making the TV screen gesture for a potential review. However, there is room for misinterpretation of these signals, especially in high-pressure situations. On-field officials may misinterpret signals or fail to recognize them, leading to confusion or missed opportunities for VAR intervention.

Moreover, VAR decisions can involve subjective judgment, such as determining the severity of a foul or interpreting the intention behind an action. Communication between VAR referees and on-field officials is crucial to ensuring a shared understanding of the incident and the reasoning behind the decision. However, differing interpretations or perspectives can still arise, leading to disagreements or inconsistent application of VAR [3].

Furthermore, communication between VAR referees and on-field officials is not directly accessible to spectators, which can sometimes create confusion or frustration among fans. They may not understand why certain decisions were made or the reasoning behind them, leading to a perceived lack of transparency. Enhancing transparency by providing clearer explanations or publicizing audio communication can help address these concerns.

Efforts are being made to address these communication challenges. Referee training programs often include modules on effective communication, and technological improvements are continually being sought to enhance the efficiency and clarity of communication systems used by VAR referees and on-field officials. The aim is to foster a seamless and transparent communication process that supports accurate decision-making while minimizing disruptions to the game.

## 5. When Philosophy Meets Technology: Immanuel Kant and VAR

Immanuel Kant, an influential German philosopher (1724–1804) of the 18th century, did not specifically discuss video assistant referees (VAR) in soccer, as certainly the technology did not exist during his lifetime. However, we can attempt to speculate on how Kant’s philosophical principles might apply to VAR.

Kant shaped Kantian ethics, a deontological ethical theory asserting that actions achieve moral merit only when driven by a sense of duty and can be universally applied without contradiction. Central to Kant’s philosophy is the categorical imperative, insisting that actions are permissible if they can be extended to all individuals without inconsistency. In accordance with Kant’s frequent assertion, our era is characterized as the age of criticism. By “criticism,” he refers to a philosophical approach that, prior to affirming, carefully evaluates, and before presuming knowledge, investigates the conditions of knowledge. Kant’s philosophy is critical not only in this broad sense but also in a specific sense as it constitutes a theory of ideas. It distinguishes itself from the extreme theories of Leibniz and Locke by discerning the formation of ideas between the outcomes of sensation and the outcomes of the spontaneous activity of pure reason. It aligns with sensationalism in recognizing that the substance of our ideas is supplied by the senses. It also aligns with idealism, asserting that their structure is the result of reason’s work. Reason, governed by its own laws, transforms the given manifold of sensation into ideas [18].

Indeed, one of the key principles he advocated was the concept of treating individuals as ends in themselves, rather than as mere means to an end. Applying this principle to the context of soccer and VAR, Kant might argue that the focus should be on the fair treatment of all players and teams involved.

From this perspective, Kant might appreciate the use of VAR in soccer, as it aims to ensure fairness and accuracy in the game. VAR technology allows referees to review critical decisions, such as goals, penalties, and red cards, by consulting video footage. This helps in minimizing human error and potential biases in officiating, thereby promoting fairness in the sport [19,20].

However, there might be some considerations from Kant’s philosophy that could raise questions or concerns about VAR. Kant emphasized the importance of human autonomy and rationality. He believed that humans possess the capacity to make moral judgments and should rely on their own rationality to determine right from wrong. In the case of VAR, the reliance on technology and external video footage could be seen as a potential encroachment on the autonomy and judgment of the referees.

Additionally, Kant valued the concept of moral responsibility. Referees play a significant role in making decisions on the field, and VAR can potentially shift some of that responsibility to technology and external video review. Kant might argue that referees should bear moral responsibility for their judgments, and the introduction of VAR could undermine this responsibility by reducing their agency.

Still, while Immanuel Kant did not specifically address video assistant referees in soccer, his philosophical principles could support the idea of the VAR as a means to promote fairness and accuracy in the game. However, concerns related to human autonomy and moral responsibility might also arise when considering the impact of VARs on the role of referees.

Thus, the implementation of Video Assistant Referees (VARs) in soccer raises several philosophical characteristics related to the intersection of technology and society. Here are some key philosophical considerations:

VARs challenge our understanding of knowledge and truth in the context of refereeing decisions. It introduces a new source of information (video replays) that can potentially override the judgments made by the on-field referee. This raises questions about the nature of expertise and the extent to which technology can enhance our understanding of the game. It also raises philosophical debates about the objectivity of truth and the reliability of human perception.

Nonetheless, the VAR introduces ethical dilemmas related to fairness and justice. While its goal is to minimize errors and improve accuracy, it raises questions about the balance between technical precision and the spirit of the game. Decisions made based on millimeter-level measurements can be seen as overly strict, potentially punishing players for insignificant infractions. Philosophical discussions arise regarding the importance of maintaining the integrity of the game versus accommodating the imperfections of human judgment.

Similar to what I have argued before [21,22], the use of VAR highlights the tension between human and technological authority. Traditional soccer matches relied solely on the judgment of the on-field referee, reflecting the authority of human decision-making. With VARs, the decision-making process shifts to a team of officials analyzing video footage, potentially diminishing the autonomy and expertise of the on-field referee. This raises questions about the role of humans in decision-making processes and the potential consequences of ceding authority to technology.

The VAR’s implementation reflects broader societal trends towards increased reliance on technology and data-driven decision-making. This shift raises philosophical questions about the potential consequences of technology-driven solutions. It prompts discussions about the impact on human experience, emotional engagement, and the preservation of the traditions and rituals associated with the sport. Philosophers may explore the implications of technology on social interactions, the balance between efficiency and human connection, and the potential loss of spontaneity and authenticity [23].

The VAR also introduces concerns related to epistemic injustice, which refers to the unfair distribution of knowledge and the power imbalances it creates. While VAR provides video evidence to support decision-making, not all stakeholders have equal access to this information. Fans in stadiums may not have the same level of visibility as viewers watching on television, potentially leading to disparities in understanding and perceptions of fairness. Philosophical discussions may revolve around the implications of technology exacerbating epistemic inequalities and its impact on the overall fairness of the game.

The VAR’s introduction raises questions about technological determinism, the belief that technology shapes society in predetermined ways. Critics argue that the implementation of VARs reflects an uncritical acceptance of technology’s potential to solve problems and improve outcomes. Philosophical analyses may delve into the underlying assumptions and ideologies that drive the adoption of VARs and examine the extent to which they reflect societal values and priorities.

Overall, the philosophical aspects of VARs in soccer raise questions of knowledge, ethics, authority, societal impact, epistemic injustice, and the relationship between technology and society. These discussions provide a deeper understanding of the complex dynamics that arise when technology is integrated into traditional practices and highlight the need for critical reflection on the implications of such advancements.

## 6. Final Whistle

Expanding on Immanuel Kant’s philosophy, we can explore further implications of his ethical framework regarding the use of video assistant referees (VARs) in soccer.

Kant emphasized the importance of universalizability in moral reasoning. According to Kant, moral principles must be applicable to all rational beings without contradiction. Applying this principle to VAR, Kant might argue that if the use of technology like VAR promotes fairness and accuracy in soccer, then it should be universally adopted across all matches to maintain consistency. This would prevent any arbitrary or unfair advantages that might arise from inconsistent implementation of VAR.

Moreover, Kant believed in the concept of moral duties and obligations. Referees, as moral agents, have the duty to officiate games impartially and fairly. VAR can be seen as a tool that assists referees in fulfilling this duty. It provides them with additional information and perspectives to make more informed decisions. From a Kantian perspective, the VAR could be viewed as an aid to referees’ moral duty rather than a replacement for their judgment. Referees would still bear the responsibility for making the final decision, using VARs to ensure fairness and accuracy.

On the other hand, Kant also emphasized the intrinsic value of human beings. According to his philosophy, human beings possess rational autonomy and should not be treated merely as a means to an end. In the context of soccer, this could raise concerns about the potential dehumanizing effects of VAR. Critics argue that excessive reliance on technology like the VAR might undermine the human element of the game and reduce it to a series of mechanical decisions. Kantians might caution against prioritizing technology over human judgment, emphasizing the importance of maintaining a balance between the two [23].

In summary, Kantian ethics can provide insights into the use of VARs in soccer. It suggests that the VAR can be justified if it promotes fairness and accuracy, aligning with the moral duties of referees. However, consideration should also be given to the potential dehumanizing effects and the need to maintain the value of human judgment in the game. Striking a balance between technology and human agency would be crucial for upholding the ethical principles associated with VAR implementation in soccer.

Similar to Zglinski’s [4] view, the success of technological aids like VAR in football is intricately tied to the underlying norms they are intended to adjudicate. Similar to legal norms, football laws can be classified into two categories: rules and standards. While VAR can greatly contribute to enforcing more explicit and specific rules, its effectiveness in policing the subjective and interpretive standards is comparatively limited in terms of the added value it provides.

On more practical grounds, it is crucial, as suggested by Zglinski [4], to consider not only the supply (i.e., regulators, YG) side but also the demand side (i.e., fans, YG) when evaluating the effectiveness of technological aids like VARs. While the design and setup of these aids are significant, equal importance should be placed on the specific tasks they are intended to fulfill. When defining the capabilities of devices like VARs, it becomes essential to take into account the context in which they are implemented and the norms they are expected to uphold.

## Data Availability

Not applicable.

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
