# Peer review of "From Sport Psychology to Action Philosophy: Immanuel Kant and the Case of Video Assistant Referees"

_behavsci, 2024, doi:10.3390/bs14040291_

Round 1
Reviewer 1 Report
Comments and Suggestions for Authors
Dear Authors,
Thank you for your manuscript. It is exciting and original. Comparing Kant's philosophy to VAR management is original and fascinating. Here are some suggestions to improve the power of your manuscript:
Firstly, I suggest you enhance the analysis of the use of VAR. For example, how many errors (in percentage) have been corrected using VAR since it was adopted? How many doubtful situations were solved with his use? But also, how did the referee perceive their decision knowing that there is VAR that can change their choices? Etc. Please try to investigate this aspect, which can give more power to your manuscript. Secondly, attempt to point out the referees' decisional process. How do referees decide? What are the mechanisms that regulate referees' decisions? About this topic, I can suggest an article that talks about coaches.' perceptions of their players. Even if referees are not coaches, this article may inspire you in improving the manuscript.
Abate Daga F, Veglio F, Cherasco GM, Agostino S. The Influence of Subjective Perceptions and the Efficacy of Objective Evaluation in Soccer School Players' Classification: A Cross-Sectional Study. Children (Basel). 2023 Apr 23;10(5):767. doi: 10.3390/children10050767. PMID: 37238315; PMCID: PMC10217720.
Finally, when you explain Kant's approach, please insert a brief recap of Kant's thinking even if Kant is a famous philosopher, maybe not all potential readers remember him and his thinking. Thus, a little recap of what he produced during his intellectual lifetime can be helpful for all readers.
Author Response
Thanks for your insights and please see my new version with TC. I have enhanced the analysis of the use of VAR. Have included how many errors (in percentage) have been corrected using VAR since it was adopted; How many doubtful situations were solved with his use. Secondly, I have attempt to point out the referees' decisional process. How do referees decide; What are the mechanisms that regulate referees' decisions. Finally, when explained Kant's approach, I have inserted a brief recap of Kant's thinking.

Reviewer 2 Report
Comments and Suggestions for Authors
The author does a great job in exploring this topic from a new prospective. The relevance of the issue is well established.
However, I'm missing the scientific value of the manuscript. It feels like a collection of opinions, but sometimes it lacks critical scientific reasoning.
Especially in the domain of decision-making there is a lot of scientific literature to explore and add to this paper, I think the work would benefit a lot from adding this prospective.
It's also missing some references.
Overall, the take is original and well thought, the issue is interesting, but I think it needs to be improved with some scientific evidence.
Maybe add some work that use behavioral tasks with decision-making (so we can really understand the process we are taking into account) would improve the scientific value of the paper.
Author Response
Thanks for your insights and please see my new version with TC.
Per your suggestion, I have added some work that use behavioral tasks with decision-making.
Added missing refs.

Reviewer 3 Report
Comments and Suggestions for Authors
I hope this letter finds you well. I had the opportunity to review your article titled, “From psychology of sport to action philosophy Immanuel Kant and the case of video assisting referees (VAR)”, which was submitted to Behavioral Sciences journal.
It is believed that the introduction of VAR in soccer is approached through Kantian theory as a very new and interesting topic. However, even in literature research, it is necessary to divide it into research methods, results, and discussion sections.
And, this study is not suitable for academic conference format.
Abstract
As the most: Modify it to As the most.
Research methods and discussion content are lacking.
As the most: Modify it to As the most.
Introduction
In the introduction, it is common to describe the necessity based on the theoretical background rather than the researcher's argument.
Please logically present and emphasize the need for research.
Research methods, research results, and discussions are somewhat lacking.
​ This study is considered to be a literature study.
Even if it is a literature study, it is necessary to explain the approach to applying Kant's theory.
The introduction of VAR in soccer is being applied in a timely manner by the International Football Association.
It seems that the need to interpret it in terms of Kant's theory is somewhat lacking.
Although the paper as a whole explains the researcher's argument well in short sentences, the research method, results, and discussion sections are judged to be somewhat lacking.
Author Response
Thanks for your insights and please see my new version with TC.
Have added a whole para on Kant's theory per you suggestion.

Round 2
Reviewer 2 Report
Comments and Suggestions for Authors
The authors improved the manuscript according to revisions
Author Response
Many thanks!
Reviewer 3 Report
Comments and Suggestions for Authors
Regarding the first revision, it is judged that the researcher made many revisions to the paper.
This reviewer believes that research without a research method is not logical.
This paper does not follow the academic format. In particular, the method of presenting references in the text was wrong. The references section is also different from the academic format.
This study consists of the researcher's claims. It is judged that there is insufficient prior research to support the researcher's claims.
The researcher is expected to present the research method and logically describe the researcher's claims based on prior research in each section.
Author Response
Thanks for your comment. This is a COMMENTERY article which does not have a research method. As you know, the goal of publishing commentaries is to advance the research field by providing a forum for varying perspectives on a certain topic under consideration in the journal. The author of a commentary probably has in-depth knowledge of the topic and is eager to present a new and/or unique viewpoint on existing problems, fundamental concepts, or prevalent notions, or wants to discuss the implications of a newly implemented innovation. A commentary may also draw attention to current advances and speculate on future directions of a certain topic, and may include original data as well as state a personal opinion.